# A Solar Array Temperature Multivariate Trend Forecasting Method Based on the CA-PatchTST Model

**DOI:** 10.3390/s25237199

**Published:** 2025-11-25

**Authors:** Yunhai Wang, Xiaoran Shi, Zhenxi Zhang, Feng Zhou

**Affiliations:** 1Key Laboratory of Electronic Information Countermeasure and Simulation Technology of Ministry of Education, Xidian University, Xi’an 710071, China; wangyunhai@stu.xidian.edu.cn (Y.W.); zhangzhenxi@xidian.edu.cn (Z.Z.); 2School of Aerospace Science and Technology, Xidian University, Xi’an 710071, China; fzhou@mail.xidian.edu.cn

**Keywords:** solar array, satellite telemetry data, temperature multivariate trend forecasting, PatchTST, cross-attention mechanism

## Abstract

System reliability, which is essential for the normal operation of satellites in orbit, is decisively governed by the performance of solar array, making accurate temperature forecasting of solar array imperative. Reliable solar array temperature forecasting is essential for predictive maintenance and autonomous power-system management. Forecasting relies on temperature telemetry data, which provide comprehensive thermal information. This task remains challenging due to the high-dimensional, long-horizon temperature sequences with inherent cross-variable coupling, whose dynamics exhibit nonlinear and non-stationary behaviors owing to orbital transitions and varying operational modes. In this context, multi-step forecasting is essential, as it better characterizes long-term dynamics of temperature and provides forward-looking trends that are beyond the capability of single-step forecasting. To tackle these issues, we propose a solar array temperature multivariate trend forecasting method based on Cross-Attention Patch Time Series Transformer (CA-PatchTST). Specifically, we decompose temperature variables into trend and residual components using a moving average filter to suppress noise and highlight the dominant component. In addition, the PatchTST model extracts local features and long-term dependencies of the trend and residual components separately through the patching encoders and channel-independent mechanisms. The cross-attention mechanism is designed to capture the correlation between temperature variables of different devices in solar array. Extensive experiments on the real solar array temperature dataset demonstrate that the CA-PatchTST surpasses mainstream baselines in root mean square error (RMSE), mean absolute error (MAE), and mean absolute percentage error (MAPE), with ablation studies further confirming the complementary roles of sequence decomposition and cross-attention.

## 1. Introduction

In recent years, communication satellite constellation technology has developed rapidly. Leveraging its capabilities for large-scale deployment and collaborative operation, such constellations provide crucial support for achieving global coverage and low-latency communication services, offering significant strategic and commercial value. However, satellites operate in harsh space environments over extended periods, where they are continuously exposed to challenges such as space radiation, debris impacts, and extreme temperature fluctuations [1,2]. Furthermore, the growing complexity of their systems significantly elevates the risk of on-orbit failures [3]. Statistical analyses reveal that power system failures account for the largest share of satellite malfunctions, with anomalies related to solar array comprising about 42% of all power system failures [4]. Excessive heating or frequent alternating thermal cycles can accelerate material aging and performance degradation of solar array, which in turn affects the stability of the power supply and threatens overall mission reliability of the satellite [5]. As the primary source of onboard power, the operational condition of solar array is thus a decisive factor in ensuring the normal operation and successful execution of satellite functions.

Given this critical role, accurate trend forecasting of multivariate telemetry parameters characterizing the thermal state of solar array has become an indispensable technology for enabling the dynamic health management of satellite constellations, enhancing autonomous operational capabilities, and extending the effective service life of spacecraft [6]. High-precision forecasting enables early detection of incipient performance degradation, supports predictive maintenance, and allows for proactive operational adjustments that can prevent cascading failures [7].

However, achieving such predictive accuracy is far from trivial. Satellite telemetry data are characterized by high dimensionality, long temporal sequences, and complex coupling relationships among parameters [8]. Moreover, the temperature variation patterns of solar array are influenced by a combination of factors such as fluctuating space environments, the periodic orbital motion of satellites, and changes in the operating modes of onboard equipment. This interplay introduces strong nonlinearity and non-stationarity into the temperature data, making straightforward modeling approaches inadequate for capturing the underlying patterns [9,10]. Current methods commonly assume reliable priors or stationarity and often degrade under orbital-phase transitions. Moreover, data-driven approaches struggle with long-horizon robustness and cross-variable coupling amid noise, missingness, and phase-dependent shifts.

In addition, the temperature series are periodic and intricately coupled across devices, thus single-step forecasting is too myopic for health management and preventive maintenance [11]. By contrast, multi-step forecasting is expected to reveal long-term trends and phase-dependent behaviors, yet existing approaches often degrade over extended horizons due to error accumulation and exposure bias in recursive decoding, and distributional shifts across orbital day–night phases, and under-modeled cross-variable coupling among structural groups. Methods built on stationarity assumptions or pointwise attention further struggle with non-stationary, multi-scale dynamics. These limitations manifest as lag across structurally coupled sensors and cumulative drift over long horizons, undermining the reliability required for on-orbit health management.

To achieve high-precision multi-step forecasts, this paper proposes a method that learns intrinsic patterns of long-term solar array temperature data and captures cross-variable dynamic correlations. In response to the above issues, we propose a method of solar array temperature multivariate trend forecasting based on CA-PatchTST, which mitigates long-horizon non-stationarity, models dynamic cross-variable coupling, and captures long-range temporal dependencies. The main contributions of this paper are outlined as follows:A patch-based, channel-independent PatchTST encoder further enhances local pattern extraction, lowers computational cost, and preserves long-range temporal patterns;Cross-attention across devices captures inter-variable correlations and enables complementary information exchange among multivariate telemetry channels;Extensive experiments on real-world GOCE satellite temperature telemetry data validate CA-PatchTST across multiple forecast horizons. The results show consistent and significant improvements over state-of-the-art baselines, with superior performance in RMSE, MAE, and MAPE, underscoring the model’s accuracy, robustness, and practical utility.

The remainder of this paper is organized as follows. Section 2 provides a systematic review and analysis of existing time-series forecasting methods, highlighting their strengths and limitations in handling satellite temperature data, and laying the theoretical foundation for the proposed model. Section 3 briefly introduces solar array temperature telemetry data, the proposed CA-PatchTST for the solar array temperature multivariate trend forecasting. Section 4 reports dataset introduction, parameter settings and experiment analysis. Section 5 provides a summary of this paper, presents the main conclusion, and offers insights for future research.

## 2. Related Work

### 2.1. Traditional Time-Series Trend Forecasting Methods

Time-series trend forecasting is critical for infrastructure management, enabling proactive maintenance and operational optimization. In satellite systems, specifically, predicting the condition of subsystems like solar array is vital for preventing performance degradation [12], yet the high dimensionality, strong inter-variable coupling, and long-range dependencies in telemetry data make the selection of appropriate methods uniquely challenging [13]. Existing trend forecasting techniques for time-series data are typically grouped into three categories: physical modeling approaches, statistical models, and data-driven models.

Physical modeling methods construct mathematical representations grounded in domain-specific knowledge and are best suited to systems whose governing laws are clearly defined and tractable. These methods are suitable for systems with well-defined physical relationships and relatively straightforward modeling processes. Haupt et al. [14] proposed a probabilistic forecasting framework that integrates physics-based modeling with machine learning techniques. In the context of renewable energy generation predicting, the authors effectively enhanced the robustness and generalization capability of predictions by combining physical models with algorithms such as random forests. Mackey and Kulikov [15] proposed a spacecraft telemetry forecasting method that leverages physics-based simulation as the primary predictor and enhances its accuracy by modeling residuals with autoregressive techniques and applying data-driven transformations, enabling more reliable real-time forecasting under uncertain system behavior. However, the highly complex, nonlinear, and strongly coupled thermal dynamics of satellite solar array, influenced by orbital and environmental interactions, make it extremely difficult to derive accurate physical-based models, limiting their applicability.

Statistical model-based time-series trend forecasting analyzes the probabilistic and statistical characteristics of systems, offering relatively high interpretability. The Autoregressive Moving Average (ARMA) model is a classic and well-established approach for time-series analysis [16], modeling stationary data through autoregressive and moving average components. Zhang et al. [17] incorporated Bayesian statistical principles into the robust fitting process of the ARMA model, thereby enhancing its stability and accuracy in satellite clock bias prediction. In addition, Long et al. [18] adopted the Prophet model, which decomposes time-series into trend, seasonality, and holiday effects, for medium-term and long-term electricity load forecasting. This approach improves modeling flexibility while maintaining good interpretability. However, these methods rely on linearity and stationarity assumptions, which prevent them from effectively capturing the pronounced nonlinear, non-stationary, and time-varying dependencies of solar array temperatures under variable operational conditions, resulting in reduced predictive accuracy.

### 2.2. Time-Series Trend Forecasting Methods Based on Deep Learning

In recent years, driven by advancements in deep learning, data-driven trend forecasting methods have demonstrated significant advantages. The Gated Recurrent Unit (GRU) employs update and reset gates to control information flow, enabling efficient modeling of sequential data and the capture of long-range temporal dependencies. Cai et al. [19] employed a Residual Convolutional Neural Network–Simple Recurrent Unit (Res-CNN-SRU) model for industrial Internet intrusion detection in a gas pipeline dataset. By combining the strong local feature extraction capability of CNN with the fast recurrent modeling of the SRU, the approach achieved high detection accuracy, low false-alarm rates, and significantly reduced training time compared to other RNN-based methods. Le Guen and Thome proposed SegRNN [20], which integrates segmentation into recurrent neural networks to capture long-term dependencies in time-series. Wang et al. [21] integrated adaptive shrinkage processing with a Temporal Convolutional Network (AS-TCN) for rolling-bearing RUL prediction, using multichannel vibration signals and outperforming strong baselines on standard industrial benchmarks. Zeng et al. proposed Dlinear [22], a decomposition-based linear framework for time-series forecasting, which separates trend and seasonal components to achieve efficient and effective long-term forecasting.

### 2.3. Transformer

The Transformer architecture has emerged as a powerful framework for time-series modeling, primarily due to its self-attention mechanism, which captures dependencies across arbitrary positions in a sequence. Unlike traditional recurrent models that struggle with long-range interactions and suffer from exploding gradients, Transformer directly models global dependencies through parallelizable attention operations [23]. As a result, Transformer is especially effective for forecasting tasks that involve high-dimensional variables and extended temporal dependencies, including applications in industrial monitoring. Moreover, the flexibility of its attention mechanism enables dynamic weighting of relevant temporal features, providing strong adaptability to non-stationary and multi-scale patterns commonly present in complex real-world time-series [24].

Building on this foundation, a range of Transformer variants have been proposed to further enhance performance in specific domains. For instance, Cuéllar et al. [25] applied an explainable anomaly detection method to spacecraft telemetry data, where the model leverages transformer-based attention modules combined with feature attribution to uncover both global and local temporal dependencies. Tested on multiple mission datasets, the approach improved anomaly detection performance, revealing better interpretability and earlier fault detection compared to baseline methods. Similarly, Yang et al. [26] applied Informer to motor-bearing vibration forecasting, where the ProbSparse self-attention mechanism enables efficient modeling of long sequences with reduced computational cost. Evaluated on multiple bearing datasets, Informer achieved state-of-the-art results and demonstrated strong potential for predictive maintenance tasks.

Attention mechanisms are increasingly employed in deep learning to selectively emphasize relevant input features and capture long-range dependencies and complex interactions. They have been widely utilized across domains such as natural language processing, computer vision, and time-series forecasting. The Squeeze-and-Excitation (SE) block provides lightweight channel-wise recalibration within each variable, strengthening salient signals and suppressing noise. Qin et al. [27] integrated an SE-based channel-attention module into a CNN–GRU hybrid for short-term distribution-network load forecasting, achieving higher accuracy and more robust peak–valley tracking on real data than baseline models. The cross-attention mechanism further extends the capability of attention by enabling information exchange across different variables, which is particularly advantageous for modeling inter-variable dependencies in multivariate time-series. Zhang et al. [28] proposed Crossformer, a Transformer-based model with cross-dimension attention that effectively captures both intra-variable temporal dependencies and inter-variable correlations, achieving superior performance on multivariate forecasting tasks.

However, the aforementioned methods still face challenges in long-sequence, high-dimensional trend forecasting tasks for critical satellite components such as solar array. Statistical models are limited by linear assumptions and struggle to characterize complex abrupt changes and non-stationary features. Recurrent neural network-based forecasting models suffer from low computational efficiency and issues such as gradient explosion and vanishing gradients when applied to long sequences [29]. Although the Transformer can parallelize the modeling of global dependencies, its pointwise attention calculation neglects local features within continuous segments and incurs quadratic computational complexity [30]. Furthermore, most existing architectures focus on univariate modeling approaches and lack the capability to capture dynamic coupling relationships among multivariate features.

## 3. Methodology

Figure 1 presents the proposed multivariate trend forecasting method for solar array temperatures based on the CA-PatchTST model. It begins by characterizing solar array temperature telemetry data, which exhibit clear trends, orbital periodicity, and random fluctuations driven by factors such as orbital cycles, solar radiation, and equipment operational states. To address these mixed patterns, moving average decomposition is applied to separate the original sequences into trend and residual components, effectively isolating low-frequency trends from high-frequency fluctuations and thereby enhancing modeling stability. The PatchTST model is employed to extract local features and long-term dependencies from both components through patch-based encoding and channel-independent mechanisms. A cross-attention mechanism is introduced, where the target device sequence serves as the Query and sequences from other devices act as Key and Value, thereby capturing dynamic coupling relationships among temperature variables across different devices. Finally, the predictions from the trend and residual branches are fused via weighted summation to produce multi-step temperature forecasts. By effectively integrating data characterization, sequence decomposition, local feature extraction, and cross-variable interaction, the proposed method significantly enhances the accuracy and robustness of long-term temperature forecasting.

### 3.1. Solar Array Temperature Telemetry Data

Solar array temperature telemetry data is a key state parameter reflecting the operational status of satellite systems. Collected in real time by high-precision onboard sensors and transmitted via ground data links, the temperature series of solar array exhibit pronounced periodic fluctuations due to the repeated transition between sunlight and Earth shadow during each 90 to 100 min orbit at an altitude of 500 to 1000 km [31]. This paper focuses on the solar array system of the GOCE satellite as the research object, which incorporates 16 temperature parameters at different locations: solar wing body temperature (2 parameters: THT10000/THT10001), BMSP structure temperature (8 parameters: THT10002-THT10019), and interface bracket temperature (6 parameters: THT10008-THT10024).

As illustrated in Figure 2, the temperature curves of different GOCE solar array components clearly demonstrate this orbital periodicity. Beyond orbital effects, the data are further influenced by solar activity cycles, seasonal illumination changes, and satellite attitude maneuvers, leading to strong non-stationarity, multi-scale variations, and nonlinear behaviors. To quantify the spatiotemporal correlations observed among measurements from different sensors—which reflect both the intrinsic thermal conductivity of the array structure and the overall in-orbit thermal environment—we employ the Pearson correlation coefficient (PCC). This metric measures the linear dependence between pairs of temperature parameters [32]. For two temperature time-series X and Y, each containing n samples, the Pearson coefficient r is defined as:(1)$$r=\frac{\sum_{i=1}^{n}\left(X_{i}-\bar{X}\right)(Y_{i}-\bar{Y})}{\sqrt{\sum_{i=1}^{n}{\left(X_{i}-\bar{X}\right)}^{2}}\sqrt{\sum_{i=1}^{n}{\left(Y_{i}-\bar{Y}\right)}^{2}}},$$
where $\bar{X}$ and $\bar{Y}$ are the mean values of X and Y, respectively. The value of r ranges from −1 to 1, with values near 1 indicating strong positive linear correlation, values near −1 indicating strong negative linear correlation, and values near 0 indicating little to no linear relationship. This method is chosen for its interpretability and widespread use in quantifying linear associations in multivariate telemetry data, thereby providing a rigorous foundation for assessing thermal coupling relationships and supporting trend forecasting among interdependent sensor readings.

We first organize the multivariate solar array temperature sequences by device. Let Xt∈ℝN denote the N-channel telemetry at time t, whose channels are partitioned into three disjoint device groups.(2)I=IBMSP∪IBracket∪IBody,    Ia∩Ib=∅a≠b,
where I is the full index set of temperature channels; IBMSP, IBody, IBracket are the subsets for the BMSP, Bracket, and Body components, respectively; and Ia denotes any one of these with a∈{BMSP,Bracket,Body}. The relations indicate that these three subsets form a partition of I, their union equals the whole set and they are pairwise disjoint, so each channel belongs to exactly one component.

For each target device v∈{BMSP,Bracket,Body}, we build two multivariate subsequences: a main component formed by channels of the target device and a cross component formed by channels of the remaining devices.(3)$$X_{t}^{\left(v,\mathrm{main}\right)}=X_{t}[I_{v}],\;\;\;X_{t}^{\left(v,\mathrm{cross}\right)}=X_{t}[I\setminus I_{v}],$$
where $I_{v}$ is the index subset for device, $X_{t}^{\left(v,\mathrm{main}\right)}=X_{t}[I_{v}]$ selects the target-device channels, whereas $X_{t}^{\left(v,\mathrm{cross}\right)}=X_{t}[I\setminus I_{v}]$ selects the complementary channels.

To effectively capture these complex dynamics, the multivariate temperature sequence $X_{t}$, is decomposed into a long-term trend component τ, which embodies orbital cycles and slowly varying environmental effects, and a residual component ε, which accounts for short-term fluctuations, noise, and irregular variations:(4)$$X_{t}=\tau_{t}+\varepsilon_{t},$$

This decomposition disentangles mixed patterns, mitigates the impact of non-stationarity, and enables more accurate modeling by allowing predictive models to simultaneously learn global evolution and local perturbations of solar array temperature dynamics.

### 3.2. CA-PatchTST

This section presents the proposed CA-PatchTST model, a novel framework designed for multivariate temperature forecasting of solar array. The model integrates a PatchTST backbone for temporal feature extraction and a cross-attention mechanism that facilitates information interaction among variables. The overall architecture comprises four key components: temperature series decomposition, PatchTST-based encoding, cross-attention-driven feature fusion, and multi-step forecast generation.

#### 3.2.1. Temperature Series Decomposition

The temperature parameters of the solar array are simultaneously affected by multiple factors such as orbital period, solar radiation, and equipment operating states, exhibiting a composite characteristic profile of distinct trend, periodicity, and random fluctuations. Therefore, decomposition of the temperature sequence serves as an effective means to disentangle these mixed patterns. Through decomposition operations, slow-changing overall trends can be effectively separated from short-term fluctuations. This helps to reduce the interference of noise and non-stationarity on model training and improve prediction stability; in parallel, it enables the model to capture long-term evolution patterns and local detail changes in temperature sequences, thereby more accurately reflecting the actual temperature behavior of solar array in complex spatial environments.

The telemetry time-series is decomposed into trend and residual components through moving average filtering [33], effectively isolating low-frequency trends from high-frequency fluctuations. Given a t length multivariate temperature time-series $X_{t}={\left(X_{t}^{\left(1\right)},\ldots ,X_{t}^{\left(N\right)}\right)}^{⊤}$ with $X_{t}^{\left(d\right)}=\left[x_{1}^{\left(d\right)},x_{2}^{\left(d\right)},\ldots ,x_{t}^{\left(d\right)}\right]$, where d denotes the d-th variable in all the N temperature parameters. The trend component $\tau_{t}^{\left(d\right)}$ is calculated by applying a moving average operation with a fixed-size kernel k, which extracts the low-frequency smoothed part of the sequence and captures its long-term trend.(5)$$\tau_{t}^{\left(d\right)}=\frac{1}{k}\sum_{j=-\left\lfloor (k-1)/2\right\rfloor }^{\left\lfloor (k-1)/2\right\rfloor }x_{i+j}^{\left(d\right)},\;\;\;\;i=1,2,\ldots ,t,$$
where j is the summation index traversing the symmetric window around, $x_{i+j}^{\left(d\right)}$ refers to the value of the i-th sensor at time offset j within the window.

The residual component $\varepsilon_{t}^{\left(d\right)}$ is obtained by subtracting the trend from the original sequence, thereby highlighting short-term fluctuations and disturbances not captured by the trend.(6)$$\varepsilon_{t}^{\left(d\right)}=X_{t}^{\left(d\right)}-\tau_{t}^{\left(d\right)},$$

The fixed-size kernel k in the moving average filter was selected based on the sampling rate and the physical periodicity of the GOCE satellite. Since the temperature telemetry is sampled every 10 min and the satellite completes one orbit in approximately 90 min, one orbital cycle corresponds to about nine sampling intervals. Therefore, k=9 was adopted so that the filter window matches one complete thermal–orbital cycle. This choice allows the moving average operation to capture the slowly varying orbital trend while suppressing high-frequency fluctuations caused by short-term environmental disturbances. A window of this scale provides a balanced decomposition, ensuring that the trend component reflects the long-term orbital behavior, whereas the residual component retains finer intra-orbital variations.

#### 3.2.2. PatchTST

In the task of solar array temperature forecasting, traditional Transformer-based time-series models can capture long-term dependencies, but they are difficult to effectively extract temperature change patterns in local continuous segments, especially sensitive to noise in solar array temperature telemetry data, which undermines predictive robustness. In addition, this type of model has the problem of high computational complexity [34], especially for long-term forecasting problems. Their scalability deteriorates when applied to long-duration, multi-year satellite temperature series.

To address these issues, we tailor the PatchTST model, as shown in Figure 3, which introduces a patch segmentation strategy inspired by image processing, dividing the temperature sequence of the solar array into fixed-length local patches and using Transformer encoders for feature extraction within each patch. This method not only reduces the computational complexity of the attention mechanism and enhances the ability to capture local temperature fluctuation patterns but also maintains the ability to express global trends through feature combinations between patches. This design is particularly suitable for the periodic and local abrupt changes in temperature data of solar array, effectively improving the predictive performance and robustness of the model for long sequences.

Compared to the quadratic computational complexity $O(L^{2}d)$ inherent in the standard Transformer model when processing long sequences, PatchTST effectively reduces the computational burden through its distinctive patching strategy. Given an input sequence length L, a patch length P, and a patch stride S, the number of patches M is given by $M=\left\lfloor \frac{L-P}{S}\right\rfloor +1$. Since the self-attention mechanism operates on this sequence of patches, its computational complexity is reduced from $O(L^{2}d)$ to $O(M^{2}d)$.

The model independently encodes and predicts the trend component and the residual component. For a multivariate time-series input X, where B is the batch size, N is the number of variables, and L is the sequence length. PatchTST first divides each variable’s time-series into overlapping patches, using a sliding window. Each patch has a length of P and a stride of S, resulting in M segments after segmentation.(7)X′=unfoldX∈ℝB×N×P×M,M=L−PS+1,

Each patch is linearly projected into a dimensional latent space, followed by the addition of a learnable positional encoding, d indicates the embedding size.(8)Z=X′⋅Wp+Epos,  Z∈ℝB×N×M×d,
where Wp denotes the linear projection weight matrix, Epos represents the learnable positional encoding, and Z is the resulting latent representation. The dimension d indicates the embedding size, which defines the hidden representation dimension of each patch after projection.

PatchTST adopts a channel-independent modeling approach. Unlike traditional multivariate attention mechanisms, this method independently models each channel during the encoding stage. Each variable has its own separate patch sequence Xc′ and generates its corresponding latent representation Zc, which helps to avoid interference across variables.(9)Zc=Xc′⋅Wp+Epos,   Zc∈ℝB×M×d,

PatchTST utilizes a standard Transformer encoder to process the sequence of patches. Each encoding layer consists of multi-head self-attention [35] and a feed-forward network. For the layer, the computation is as follows:(10)$$Z^{\left(l\right)}=\mathrm{LayerNorm}\left(Z^{\left(l-1\right)}+\mathrm{MultiHeadAttn}\left(Z^{\left(l-1\right)}\right)\right),$$(11)$$Z^{\left(l\right)}=\mathrm{LayerNorm}\left(Z^{\left(l\right)}+\mathrm{FFN}\left(Z^{\left(l\right)}\right)\right),$$

#### 3.2.3. Cross-Attention Mechanism

Although the PatchTST model can effectively extract time-series features of each individual variable, its modeling ability is limited when dealing with multi-source variables and auxiliary sequences. To fully capture the coupling relationship within multivariate temperature sequences, we introduce a cross-attention mechanism on the basis of the device grouping described in Section 3.2.1, to enhance feature complementarity across sequences.

For a target device group V(t), we define its multivariate time-series as the main component, denoted as Zmain∈ℝB×|V(t)|×M×d, which serves as the primary sequence for prediction. The multivariate time-series from all other device groups, V(c), are concatenated to form the cross component, denoted as Zcross∈ℝB×|V(c)|×M×d, which provides contextual information. V(t) and V(c) denote the number of temperature channels in the target group and the combined cross groups, respectively. This design explicitly models the target device’s response to the thermal state of the entire solar array system.

The PatchTST encoder first processes the main and cross components independently, yielding their respective patch-based feature representations $H_{\mathrm{main}}\;$ and $H_{\mathrm{cross}}$. In the cross-attention layer, the Query (Q) is derived exclusively from the main component’s representation $H_{\mathrm{main}}\;$, compelling the model to focus on and refine the prediction of the target device. Conversely, the Key (K) and Value (V) are projected from the cross component’s representation $H_{\mathrm{cross}}$, which encapsulates the thermal dynamics of the auxiliary devices. 

The cross-attention layer consists of multi-head cross-attention and a feed-forward network, as shown in Figure 4. The main sequence undergoes a linear transformation to generate the Query, while the cross sequences undergo linear transformations to generate the Key and Value. $W_{q}$, $W_{k}$ and $W_{v}$ are learnable projection matrices.(12)$$Q=H_{\mathrm{main}}\;W_{q},\;K=H_{\mathrm{cross}}W_{k},\;V=H_{\mathrm{cross}}W_{v}$$

The cross-attention mechanism computes the relevance between Query and Key, and uses the resulting attention distribution to adaptively aggregate features from Value.(13)$$\mathrm{CrossAttention}(Q,K,V)=\mathrm{softmax}\left(\frac{QK^{T}}{\sqrt{d_{k}}}\right)V,$$

By computing attention weights between Query (target device) and Key (auxiliary devices), the model can adaptively capture coupling relationships among temperature variables across different devices. This mechanism enables the model to integrate cross-device contextual information, thereby achieving a more comprehensive perception of the system state and improving the accuracy and robustness of overall temperature trend forecasting and local dynamics of the solar array.

The final output of the cross-attention layer $Z_{\mathrm{out}}$ is fused with the original input Q through a residual connection and layer normalization, followed by a feed-forward neural network:(14)$$Z_{1}=\mathrm{LayerNorm}\left(Q+\mathrm{MultiHeadAttn}\left(Q,K,V\right)\right),$$(15)$$Z_{2}=\mathrm{LayerNorm}\left(Z_{1}+\mathrm{FFN}\left(Z_{1}\right)\right),$$(16)$$Z_{\mathrm{out}}=\mathrm{CrossAttentionLayer}\;\left(Z_{Main},Z_{\mathrm{Cross}}\right),$$

#### 3.2.4. Output Fusion

The merge module converts branch features into horizon-wise forecasts and fuses the trend and residual components. The hidden states from the trend and residual branches, denoted as Ztrend,Zres∈ℝB×N×M×d, are first flattened along the last two dimensions per variable to obtain Htrend,Hres∈ℝB×N×(Pd). Each variable c is then independently projected to the forecast horizon H via a dedicated linear layer.(17)Y^ctrend=HctrendWctrend+bctrend,Y^cres=HcresWcres+bcres,

The final multi-step forecast is obtained through an element-wise summation of the trend and residual components. This additive fusion combines the long-term evolutionary patterns captured by the trend component with the short-term fluctuations modeled by the residual component, yielding a comprehensive forecast.(18)Y^=Y^trend+Y^res∈ℝB×N×H,

We used the mean squared error (MSE) as the loss function to train our forecasting model due to its stable gradient behavior and suitability for regression on continuous telemetry. The MSE quantifies the average squared discrepancy between predictions and ground truth across all variables, time steps, and batch samples [36], providing a uniform optimization objective for both the trend and residual forecasting branches. The loss is computed as:(19)$$L_{\mathrm{MSE}}=\frac{1}{BNH}\sum_{b=1}^{B}\sum_{n=1}^{N}\sum_{t=1}^{H}{\left(y_{b,n,t}-{\hat{y}}_{b,n,t}\right)}^{2},$$
where $y_{b,n,t}$ denotes the ground-truth value, and ${\hat{y}}_{b,n,t}$ the predicted value. This formulation ensures balanced gradient propagation, promotes smooth training dynamics, and facilitates the simultaneous optimization of both components in our architecture.

Back-propagation benefits from the additive form of the final forecast. With $\hat{y}=\hat{\tau }+\hat{\varepsilon }$, the loss gradient decomposes cleanly so that each branch receives gradient signals consistent with the overall forecast.(20)$$\frac{\partial L_{\mathrm{MSE}}}{\partial \hat{y}}=\frac{\partial L_{\mathrm{MSE}}}{\partial \hat{\tau }}=\frac{\partial L_{\mathrm{MSE}}}{\partial \hat{\varepsilon }},$$

This symmetry yields balanced updates for the trend and residual heads and drives consistent learning through subsequent layers. Model parameters θ are optimized with the Adam optimizer.(21)$$\theta \leftarrow \theta -\eta \nabla_{\theta }L_{\mathrm{MSE}},$$

#### 3.2.5. Model Architecture

The overall architecture of the proposed CA-PatchTST model is summarized in Table 1, which outlines the hierarchical structure, key operations, and the corresponding input/output shapes of each component. The model is designed as a dual-branch pipeline that processes multivariate temperature sequences through four major stages: decomposition, patch-based encoding, cross-attention fusion, and multi-step forecast output.

The multivariate input is decomposed by a moving-average filter into trend and residual components, each processed by an independent PatchTST branch that patchifies channels, applies linear projection with positional encodings, and uses channel-independent Transformer encoders. A cross-attention module fuses information across structural device groups (target as Query, auxiliaries as Key/Value), with output refined by residual connections, LayerNorm, and FFN. Prediction heads map both branches to multi-step forecasts, combined element-wise to yield final temperatures; trained end-to-end with MSE, the additive fusion ensures balanced optimization, while the design remains scalable and interpretable. Table 1 provides detailed component descriptions and tensor shapes for reproducibility.

#### 3.2.6. Implementation Procedure of CA-PatchTST

To provide a clear and actionable overview of the proposed methodology, Algorithm 1 presents the complete operational workflow of the CA-PatchTST model in pseudocode. It outlines the step-by-step computational process, from the initial organization of input sequences and their decomposition into trend and residual components, through the core stages of patch-based encoding, cross-attention fusion, and channel-independent transformation, to the final generation of multi-step forecasts. This procedural blueprint is designed to facilitate a straightforward and accurate implementation of the model.

**Algorithm 1.** CA-PatchTST for Solar Array Temperature Trend Forecasting1:  Input: Multivariate temperature sequence: X∈ℝB×N×L,         Device partition: {GBMSP,GBracket,GBody},       
patch length P, stride S, forecast horizon H 2: Organize sequences by device groups: Xmin=X[:,Gtarget,:],Xcross=X[:,G∖Gtarget,:]3: for component c∈{main, cross} do: 4:       Apply moving average filtering: Xtrend(c)=MA(X(c))5:       Compute residual component: Xres(c)=X(c)−Xtrend(c)6: end for7: for component c∈{trend, res} do: 8:       for branch b∈{main, cross} do: 9:               Patching: Pc(b)=Unfold(Xc(b),kernel=P,stride=S)∈ℝB×Nb×M×P10:                Linear Projection & Position Encoding: Zc(b)=Pc(b)Wp+Epos∈ℝB×Nb×M×D11:                Channel-independent TST Encoding: Zc(b)=TSTEncoder(Zc(b))12:       end for13: end for14: for component c∈{trend, res} do: 15:       Extract representations: Zmain=Zc(main),Zcross=Zc(cross)16:       Multi-head Cross-Attention:                       Q=ZmainWQ,K=ZcrossWK,V=ZcrossWV, Attention=SoftmaxQK⊤dkV17:       Residual Connection & Layer Normalization: Z′=LayerNorm(Attention+Zmain)18:       Feed-Forward Network: Zcout=LayerNorm(FFN(Z′)+Z′)19: end for20: for component c∈{trend, res} do: 21:       Flatten and project to forecast horizon: Y^c=Flatten(Zcout)Wo∈ℝB×Ntarget×H22: end for23: Additive Fusion: Y^=Y^trend+Y^res 24: Compute MSE loss: L=1B⋅Ntarget⋅H∑‖Y−Y^‖225: Update parameters: θ←θ−η∇θL26: Output: Multi-step forecasts Y^∈ℝB×N×H 

## 4. Experiments

### 4.1. GOCE Satellite Temperature Dataset

The GOCE satellite is a scientific mission satellite of the European Space Agency (ESA), operating in a low Earth orbit at approximately 263 km with an orbital period of 90 min [37]. This dataset covers multivariate temperature telemetry data of GOCE satellite solar array from March 2009 to June 2012, using ESA’s recommended 10 min resampled data, which has undergone prior calibration and statistical processes by ESA to address missing values and irregular sampling rates.

The dataset has significant orbital period characteristics: every 90 min of orbital period, the satellite experiences about 60 min of sunlight exposure and 30 min of Earth’s shadow period, causing periodic temperature fluctuations [38], as Figure 5 shows. The +Z plane mainly faces the Sun with the largest temperature variation amplitude, whereas the—Z plane faces away from the Sun with relatively stable temperature. 16 temperature sensors at different spatial positions provide multidimensional spatial temperature distribution and thermal gradient information of solar array. This multivariate temperature dataset has strong periodic patterns, multi-level structural information and characteristics of an extreme thermal environment, providing a clear pattern basis for local feature extraction.

The Pearson correlation coefficient is employed to quantify the linear correlations among temperature parameters across the solar array’s structural components, revealing a clear hierarchy of thermal interdependencies. Analysis demonstrates exceptionally strong intra-group cohesion, particularly within the Interface Bracket group where sensors such as THT10008, THT10022 and THT10024 exhibit near-unity correlation coefficients of 1.00, indicating virtually identical thermal profiles due to spatial proximity and shared thermal mass. The matrix further reveals distinct inter-group coupling patterns, with bracket sensor THT10002 showing a strong correlation of 0.73 with specific BMSP sensors. The consistent weak positive correlations between Interface Bracket and Wing Body sensors, exemplified by THT10000 at 0.25, demonstrate additional thermal connectivity. This correlation matrix provides a crucial physical prior that not only demonstrates the array’s operation as a complex interconnected thermal system but also establishes a quantitative benchmark for validating whether our CA-PatchTST model’s learned attention patterns accurately reflect these measurable physical relationships [39].

### 4.2. Experiment Settings

To validate the effectiveness of the proposed method for solar-array temperature forecasting, we used the GOCE satellite temperature dataset with a 7:2:1 split for training, validation, and testing. The lookback window was set to 144 time steps, which equals 24 h at 10 min sampling. This duration spans a complete daily cycle, enabling the model to learn both intra-orbit illumination/eclipse transitions and diurnal thermal variation. Forecast horizons were set to 144, 432, and 720 steps, corresponding to 24, 72, and 120 h. These three horizons provide a balanced evaluation of short-, medium-, and long-range prediction scenarios that are relevant to operational planning. The model configuration is summarized in Table 2, where a learning rate of 0.0001 was used to ensure stable gradient updates and a dropout rate of 0.05 is adopted to mitigate overfitting.

All experiments were conducted on a workstation with an NVIDIA RTX 4060 GPU from NVIDIA Corporation, (Santa Clara, CA, USA) an Intel Core i7-14650HX CPU from Intel Corporation, (Santa Clara, CA, USA) and 32 GB of RAM, using Python 3.8, PyTorch 1.12, and CUDA 11.6. The hardware and software stack was kept fixed across all runs to maintain consistent comparisons and support reproducibility.

### 4.3. Evaluation Metrics

To evaluate the forecasting accuracy of the model, RMSE, MAE, and MAPE were used as performance metrics. Here, $y_{i}$ denotes the ground truth, ${\hat{y}}_{i}$ denotes the predicted value, n is the number of samples, and $\bar{y}$ represents the mean of the ground truth values. These evaluation metrics characterize the deviation between the predicted and actual values from different perspectives.(22)$$\mathrm{RMSE}=\sqrt{\frac{1}{n}\sum_{i=1}^{n}{\left(y_{i}-{\hat{y}}_{i}\right)}^{2}},$$(23)$$\mathrm{MAE}=\frac{1}{n}\sum_{i=1}^{n}|y_{i}-{\hat{y}}_{i}|,$$(24)$$\mathrm{MAPE}=\frac{1}{n}\sum_{t=1}^{n}\left|\frac{y_{t}-{\hat{y}}_{t}}{y_{t}}\right|\times 100\%,$$

### 4.4. Comparison with Other Methods

#### 4.4.1. Attention Visualization

To verify whether the cross-attention mechanism within the trained CA-PatchTST model learns physically meaningful relationships, we compared its learned attention patterns against the static statistical correlations inherent in the data. The objective was to examine whether that model’s dynamic, learned dependencies align with the inherent structural couplings suggested by the Pearson correlation coefficient matrix.

The analysis was performed using the final, pre-trained CA-PatchTST model in inference mode, with its weights frozen. The model was configured with a lookback window of 144 time steps (24 h), a forecast horizon of 432 time steps (72 h), and a patch length of 16. To ensure the evaluation reflects the model’s generalization capabilities, the attention weights were computed using 64 representative batches drawn from the test set. This approach provided a snapshot of the model’s learned dependencies when processing previously unseen data.

To analyze inter-group dynamics, we extracted raw attention weights from the cross-attention layers by iteratively using each structural group (Interface Bracket, BMSP Structure, Wing Body) as the Query and the other two as Key and Value, yielding three attention maps. We averaged the attention weights across samples, heads, and temporal patches, resulting in a matrix of mean attention scores for each Query–Key pair. This matrix was column-wise z-score normalized; positive values (red) indicate higher-than-average attention, while negative values (blue) indicate lower focus, reflecting relative predictive importance across parameters.

The resulting attention heatmaps exhibit strong and consistent agreement with the structural relationships captured by the PCC matrix as Figure 6, providing numerical validation that the model’s learned attention aligns with measurable physical couplings. When the BMSP Structure serves as the Query (Figure 7a), the PCC matrix shows strong correlations with the Wing Body, such as values reaching 0.73 between certain sensors, and moderate yet notable correlations with the Interface Bracket, exemplified by coefficients around 0.25. This pattern is clearly reflected in the attention heatmap through consistently high positive z-scores from the BMSP parameters toward both of the other groups. Similarly, with the Wing Body as the Query (Figure 7b), the PCC matrix confirms strong ties to the BMSP Structure, with correlations up to 0.73, but weak associations with the Interface Bracket, some near zero or slightly negative like −0.01. The cross-attention mechanism accurately mirrors this distinction with predominantly positive attention toward the BMSP Structure and significantly weaker or negative z-scores toward the Bracket group. When the Interface Bracket is used as the Query (Figure 7c) the PCC indicates moderate correlation with the BMSP Structure, for instance 0.25, and the weakest correlation with the Wing Body, around 0.10.

The consistent alignment between the learned attention patterns of the model and the static Pearson correlations across all three structural groups demonstrates that the cross-attention mechanism captures physically interpretable relationships from the time-series data. This result confirms the capacity of the model to represent the underlying structural dynamics of the system in a manner consistent with empirical correlation patterns.

#### 4.4.2. Comparison with Other Forecasting Methods

This paper systematically evaluates the performance of four advanced models, including the proposed CA-PatchTST, the classic linear model Dlinear (2022) [22], the RNN-based SegRNN (2023) [20], and the Transformer-based Informer (2021) [26], and two recently proposed Transformer variants, TimesNet (2023) [40] and iTransformer (2024) [41], across near-term, medium-term, and long-term forecast horizons of 24, 72, and 120 h for multivariate solar array temperature forecasting. As clearly demonstrated in Figure 8c, the proposed CA-PatchTST model consistently outperforms all baseline methods, achieving notably low MAPE values of 9.26% at 24 h, 13.68% at 72 h, and 24.67% at 120 h, significantly surpassing alternative models across all horizons.

This performance advantage stems from the integrated design of the model: the cross-attention mechanism effectively captures inter-variable correlations among structural groups, while patch-based encoding and trend–residual decomposition jointly enhance local feature extraction and noise suppression. The model maintains robust accuracy over extended horizons, mitigating error accumulation and temporal drift, with the performance gap particularly widening against sequence-sensitive baselines such as SegRNN, as evidenced by the comparison of 24.67% versus 47.22% MAPE at 120 h. Moreover, compared with recent Transformer variants such as TimesNet, iTransformer, and Informer, CA-PatchTST consistently achieves lower error and higher stability across all horizons, demonstrating superior capability in modeling non-stationary and long-range dependencies in multivariate satellite telemetry data. These results align with earlier ablation studies, confirming the necessity of both cross-attention and decomposition modules in sustaining prediction stability. Figure 8 visually illustrates the superior temporal generalization capability and structural effectiveness of CA-PatchTST, underscoring its suitability for reliable long-horizon satellite temperature forecasting and predictive maintenance applications.

To visually evaluate the superior performance of the proposed CA-PatchTST model in long-sequence forecasting tasks, we designed a rigorous comparative experiment. All models were evaluated under identical settings, using an input sequence of 144 time steps (24 h) to forecast the subsequent 432 time steps (72 h) in a single forward pass. Figure 9 shows the forecasting results of these models for the key sensor THT10002 on the BMSP structure during the first week of January 2012, which comprises two subplots: the upper panel compares the predicted and ground truth curves, while the lower panel displays the corresponding error distributions.

From the upper subplot, the satellite telemetry temperature series displays clear periodic fluctuations with sharp peaks and troughs, reflecting its complex, nonlinear and dynamic characteristics. Among all compared models, CA-PatchTST demonstrates the highest consistency with the true values. Its predicted curve aligns almost perfectly with the ground truth, successfully capturing both long-term periodic trends and transient variations. In contrast, models such as DLinear, SegRNN, Informer, TimesNet and iTransformer exhibit visible deviations, especially during rapid temperature transitions, indicating weaker adaptability to high-frequency and nonlinear dynamics inherent in satellite telemetry data.

The error distributions in the lower subplot further confirm the superiority of CA-PatchTST. Its error curve exhibits the smallest fluctuation range, remaining consistently close to zero, which reflects high stability and reliability throughout the forecast horizon. The error curves of the other three models, however, demonstrate considerably larger and irregular fluctuations, far exceeding those of CA-PatchTST. Notably, these baseline models produce abnormal error spikes around critical peaks and troughs, which would be unacceptable in high-precision applications such as spacecraft thermal management.

This comparative experiment provides strong evidence that the proposed CA-PatchTST model possesses significant advantages over mainstream models in handling satellite telemetry data with complex dynamics. Its high forecasting accuracy, stability, and capability to capture critical variations make it a highly suitable and promising approach for thermal analysis and forecasting in spacecraft applications.

To quantitatively assess the computational efficiency and deployment feasibility of the proposed model for resource-constrained onboard satellite systems, we conducted a comparative analysis of key efficiency metrics against several state-of-the-art baselines. The evaluation was performed under a standardized experimental setup to ensure a fair comparison: a batch size of 32, an input sequence length of 144, and a prediction horizon of 432 were applied uniformly across all models. All experiments were executed on the same hardware platform with a fixed random seed to eliminate performance variability. We report three critical metrics for each model: the number of parameters, the average inference time per batch, and the peak GPU memory consumption during inference.

As demonstrated in Table 3, the models exhibit distinct efficiency profiles corresponding to their architectural families. Among Transformer-based models, our proposed CA-PatchTST demonstrates superior efficiency, achieving the most compact architecture (4.9 M parameters), the fastest inference speed (10.5 ms), and the lowest memory consumption (2.4 GB) within its architectural class. This efficiency is attributed to the patch-based segmentation strategy and channel-independent modeling, which effectively reduces computational redundancy while preserving representational capacity. Linear models, exemplified by DLinear, achieve even higher computational efficiency due to their structural simplicity; while DLinear infers slightly faster, CA-PatchTST maintains a markedly better forecasting accuracy, justifying its marginal computational overhead. In contrast, RNN-based models like SegRNN exhibit the lowest efficiency across all metrics, which aligns with the known challenges of recurrent architectures in processing long sequences. The results indicate that CA-PatchTST strikes a favorable trade-off, delivering significantly superior forecasting accuracy over simpler linear models with only a marginal computational overhead, while simultaneously overcoming the efficiency limitations of RNN-based approaches. This makes it a practical and effective solution for deployment in resource-constrained satellite systems where both prediction performance and operational efficiency are critical.

### 4.5. Ablation Experiment

#### 4.5.1. Component Ablation Experiment

To rigorously quantify the individual contributions of the core components within the proposed CA-PatchTST framework, a comprehensive set of ablation studies was carried out. As a critical methodology in deep learning research, ablation studies systematically remove specific elements from the full model to isolate and measure their impact on performance. This approach verifies whether each module functions as intended and clarifies their complementary roles.

As detailed in Table 4, we specifically ablated the cross-attention mechanism and the sequence decomposition module to assess their individual contributions to forecasting accuracy. The results show that the complete CA-PatchTST model achieves the best performance across all forecast horizons in terms of RMSE, MAE, and MAPE, confirming its superior accuracy and robustness. Through systematic evaluation of four model variants, we further dissect the individual and combined effects of the cross-attention and decomposition components.

The CA-PatchTST model, incorporating both cross-attention and sequence decomposition modules, consistently achieves the lowest errors across all forecast horizons, particularly exemplified at the 144-step (24 h) horizon with values of RMSE of 1.538, MAE of 0.885, and MAPE of 9.26%, underscoring the synergistic effect of integrated components. Removing the cross-attention module leads to noticeable performance degradation in short-term predictions, with RMSE rising to 1.710 and MAPE increasing to 16.35% at 144 steps, highlighting its essential role in capturing inter-variable dependencies and refining cross-device interactions. Conversely, disabling the decomposition module significantly impairs medium- and long-term forecasting performance, as indicated by the increase in RMSE to 2.199 and MAPE to 19.54% at 432 steps (72 h), confirming the importance of moving-average decomposition in isolating trend–residual patterns and stabilizing long-horizon forecasts. The baseline model without both components yields the poorest performance across all horizons, especially over longer sequences where MAPE reaches 28.13% at 720 steps, affirming that neither mechanism alone suffices to model the complex, non-stationary dynamics inherent in solar array temperature data.

The consistent superiority of the full model across all horizons confirms the complementary roles of the two components: cross-attention effectively models short-term, cross-sensor thermal couplings, while the decomposition module captures underlying trend-periodicity structures essential for medium- and long-term forecasting.

#### 4.5.2. Backbone-Attention Ablation Experiment

To comprehensively evaluate the impact of different encoder architectures and attention mechanisms on model performance, we conducted a systematic ablation study comparing four representative encoder backbones (PatchTST, TCN, SRU, and iTransformer), each paired with either cross-attention or squeeze-and-excitation (SE) attention mechanisms. All models were evaluated under a forecast horizon of 432 steps (72 h) using the GOCE satellite temperature dataset, with consistent input settings including a lookback window of 144 time steps, patch length of 16, and stride of 8 where applicable. Performance was measured using RMSE, MAE, and MAPE to ensure a comprehensive assessment of forecasting accuracy.

As clearly demonstrated in Table 5, the combination of the PatchTST encoder with cross-attention significantly outperforms all other encoder–attention combinations across every metric, achieving an RMSE of 2.079, MAE of 1.231, and MAPE of 13.68%. This superior performance can be attributed to the synergistic effects of PatchTST’s patch-based segmentation and channel-independent encoding, which effectively capture both local temporal patterns and long-range dependencies, coupled with the cross-attention mechanism’s capacity to model dynamic inter-variable correlations across different structural groups of the solar array. In contrast, the same PatchTST encoder augmented with SE attention, which performs only channel-wise recalibration without explicit cross-variable interaction, produces substantially worse results, with an RMSE of 2.884 and MAPE of 22.76%, underscoring the critical importance of modeling inter-sensor dependencies in multivariate forecasting.

Other encoder architectures, including TCN, SRU, and iTransformer, consistently underperform the PatchTST-based models regardless of the attention mechanism used. For instance, the TCN encoder paired with cross-attention attains an RMSE of 3.441 and MAPE of 27.65%, while the SRU-based model reaches an RMSE of 3.385 and MAPE of 18.69%. The iTransformer model with cross-attention performs slightly better among non-PatchTST encoders, yet still falls short with an RMSE of 3.265 and MAPE of 20.56%. These results suggest that while cross-attention generally enhances each backbone’s ability to leverage cross-variable information, the architectural advantages of PatchTST, such as its patching strategy, reduced computational complexity, and improved local feature extraction, are essential for achieving state-of-the-art performance in long-horizon, multi-sensor temperature forecasting.

The consistent superiority of the CA-augmented PatchTST model affirms its efficacy in capturing both complex temporal dependencies and physically meaningful interactions among solar array temperature variables, as further corroborated by the attention alignment analysis in Section 4.4. This ablation study not only confirms the rationale underlying the proposed CA-PatchTST framework but also highlights the limitations of conventional SE-style attention and other encoder architectures in handling high-dimensional, non-stationary satellite telemetry data.

## 5. Conclusions

This paper proposes CA-PatchTST, a multivariate forecasting framework for solar array temperature, and validates its superior performance on the GOCE satellite telemetry dataset, where empirical results demonstrate consistent improvements in MAE, RMSE, and MAPE across multiple horizons, confirming the model’s accuracy and robustness for long-horizon on-orbit forecasting. The methodological contribution lies in the coherent integration of complementary modules into a unified pipeline: the moving-average decomposition, matched to the satellite’s orbital cycle, suppresses high-frequency fluctuations and stabilizes long-horizon learning; the patch-based, channel-independent encoder enhances local feature extraction and global temporal representation while improving computational efficiency; and the cross-attention mechanism captures inter-device correlations by fusing auxiliary thermal streams, with learned attention patterns aligning with physical couplings revealed by the Pearson correlation matrix. Together, these components form a compact and interpretable framework whose primary application value lies in providing critical data support for autonomous satellite operations and informed power system decision-making.

In future work, we will advance our prototype toward operational deployment by focusing on model lightweighting and online adaptation. For lightweighting, we will implement pruning, quantization, and knowledge distillation to optimize the efficiency–accuracy trade-off. For online adaptation, we will develop continual learning with experience replay and anomaly-aware updates to maintain performance under environmental shifts. These co-designed improvements will enhance robustness and enable long-term autonomous satellite power management.

## Figures and Tables

**Figure 1 sensors-25-07199-f001:**
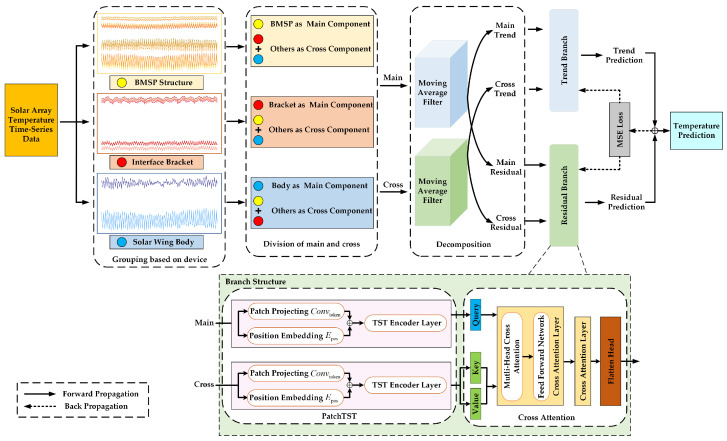
The Overall Framework of the Proposed CA-PatchTST.

**Figure 2 sensors-25-07199-f002:**
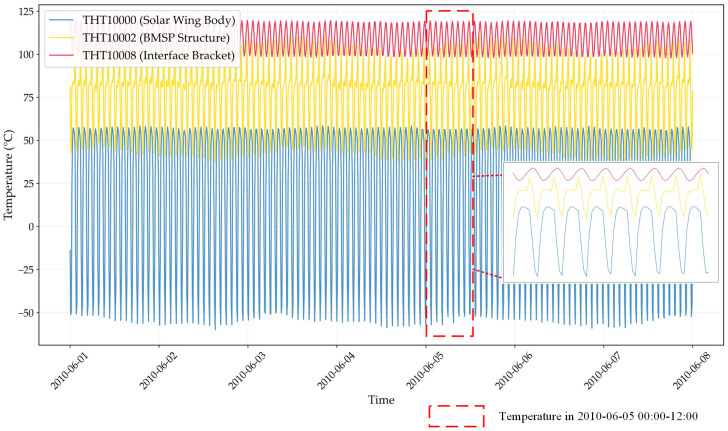
GOCE Satellite Solar Array Temperatures Trend over Time.

**Figure 3 sensors-25-07199-f003:**
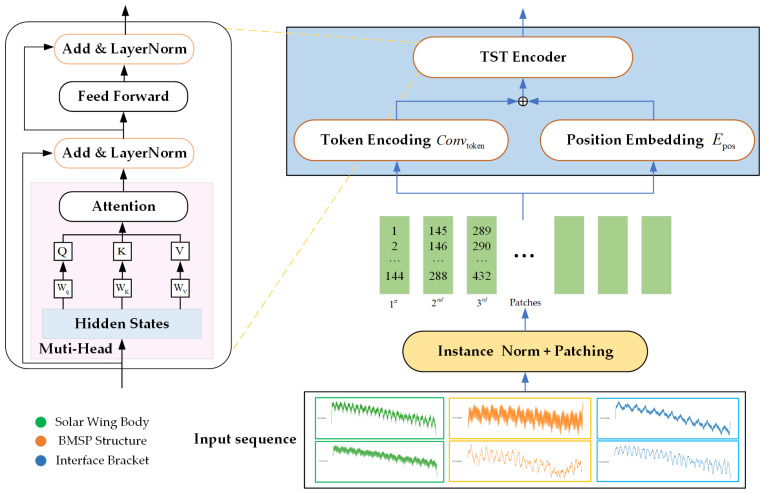
The Flowchart of the PatchTST Model.

**Figure 4 sensors-25-07199-f004:**
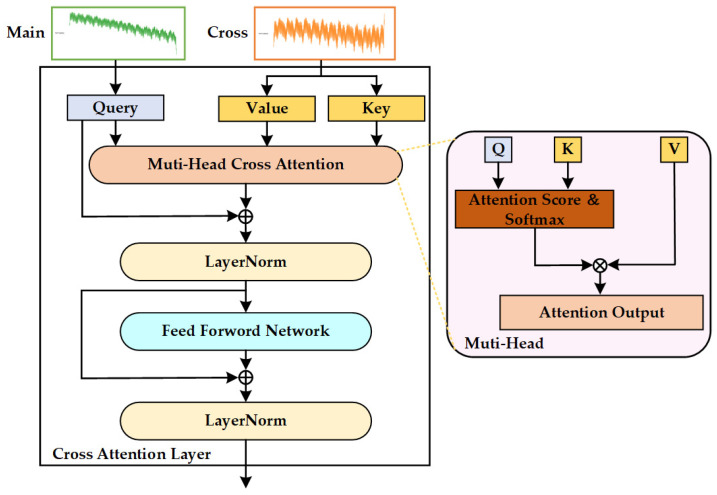
The Flowchart of Cross-attention Layer.

**Figure 5 sensors-25-07199-f005:**
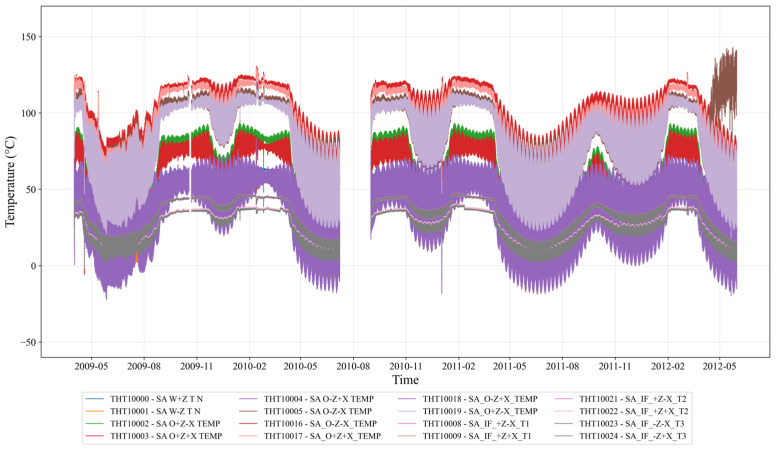
Time-Series of Temperatures for the GOCE Satellite Solar Array.

**Figure 6 sensors-25-07199-f006:**
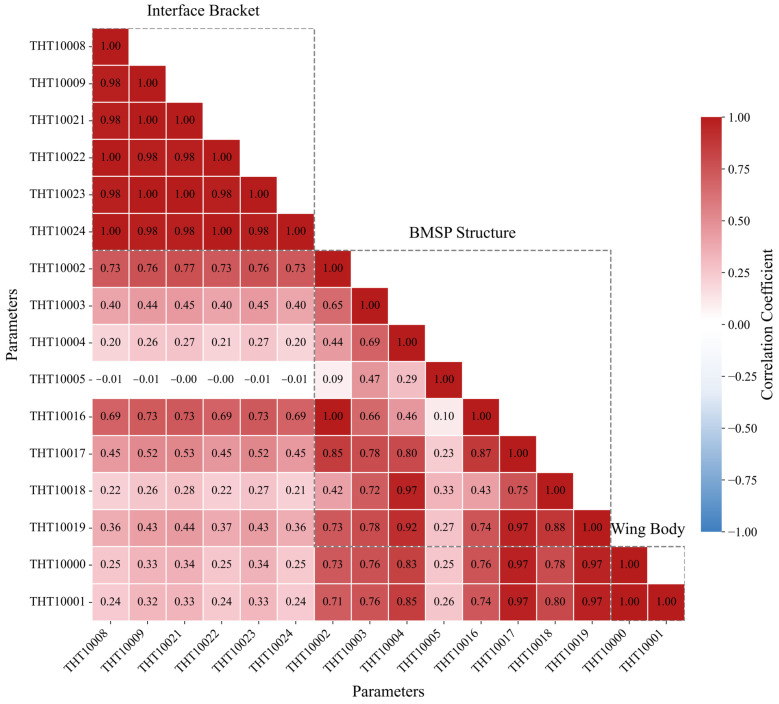
Pearson Correlation Matrix of Solar Array Temperatures.

**Figure 7 sensors-25-07199-f007:**
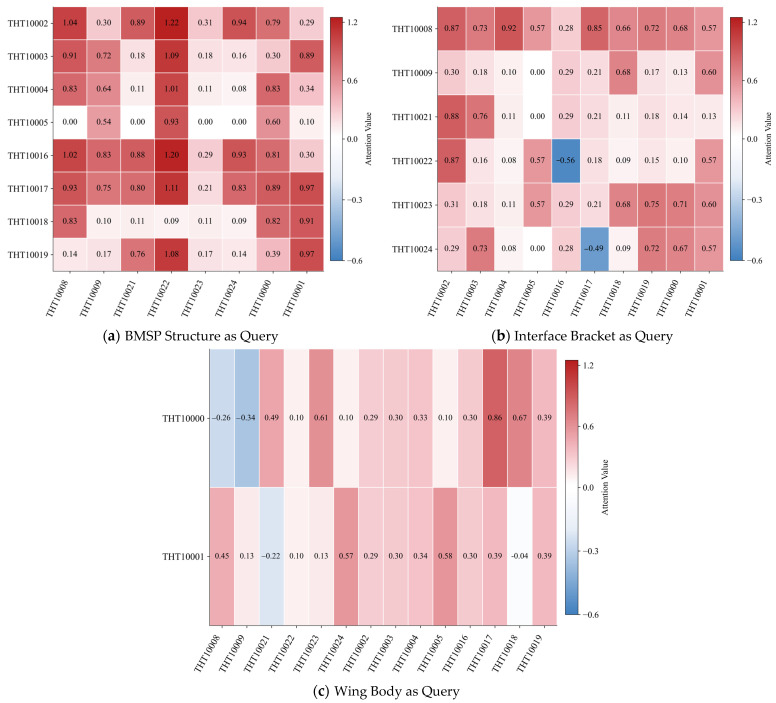
Cross-Attention Analysis by Structural Grouping.

**Figure 8 sensors-25-07199-f008:**
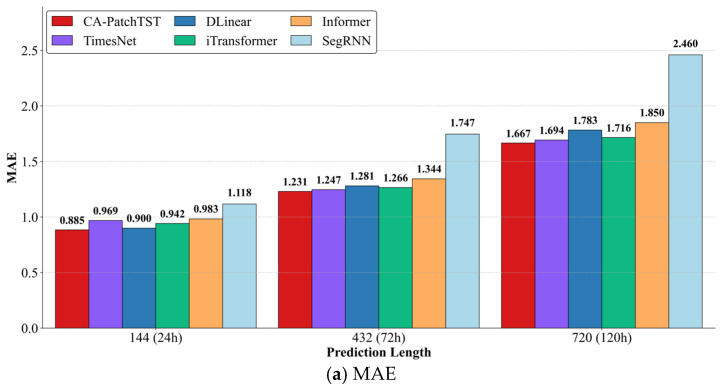
Performance Comparison of Different Models.

**Figure 9 sensors-25-07199-f009:**
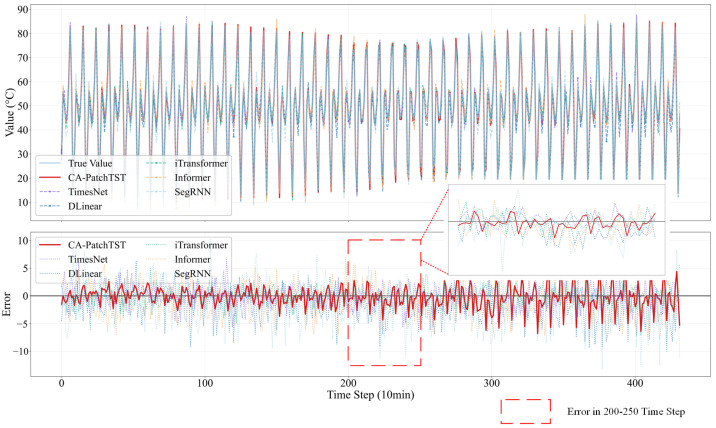
Comparative Forecasting Performance of THT10002 Parameter across Different Models.

**Table 1 sensors-25-07199-t001:** Model Structure *.

Block	Layer	Operation	Input Shape	Output Shape
Decomposition	Input	Main sequence, Cross sequence	[B, N, seq_len]	[B, $N_{\mathrm{tat}}$, seq_len] [B, $N_{\mathrm{aux}}$, seq_len]
	Moving-average decomposition	Split each sequence into trend and residual component via MA filter	[B, $N_{\mathrm{tat}}$, seq_len] [B, $N_{\mathrm{aux}}$, seq_len]	[B, $N_{\mathrm{tat}}$, seq_len] × 2 [B, $N_{\mathrm{aux}}$, seq_len] × 2 (Trend/Res.)
PatchTST (Single-Branch)	Patching	ReplicationPad1d	[B, $N_{\mathrm{sub}}$, seq_len]	[B, $N_{\mathrm{sub}}$, M, P]
		Unfold and Permute		
	Patch projection + Pos encoding	Linear projection	[B, $N_{\mathrm{sub}}$, M, P]	[B, $N_{\mathrm{sub}}$, M, D]
		Dropout		
		Add learnable positional encodings		
	TST Encoder × N (Channel-independent for per variable).	Multi-Head Self-Attention	[B, $N_{\mathrm{sub}}$, M, D]	[B, $N_{\mathrm{sub}}$, M, D]
		Dropout		
		Residual shortcut		
		LayerNorm		
		FFN: Linear (D → 2D) → GELU → Dropout → Linear (2D → D)		
		Residual shortcut		
		LayerNorm		
Cross-Attention (Single-Branch)	Pre-CA	Q from main, K/V from cross	[B, $N_{\mathrm{tat}}$, M, D] [B, $N_{\mathrm{aux}}$, M, D]	Q: [B, $N_{\mathrm{tat}}$, M, D] K/V: [B, $N_{\mathrm{aux}}$, M, D]
	Cross-Attention block × M	Multi-Head Cross-Attention: Reshape to muti-heads → Attn softmax → Dropout → Attn·V → Concat heads	[B, $N_{\mathrm{tat}}$, M, D] [B, $N_{\mathrm{aux}}$, M, D]	[B, $N_{\mathrm{tat}}$, M, D]
		Dropout		
		Residual shortcut		
		LayerNorm		
		FFN: Linear (D → 4D) → GELU → Dropout → Linear (4D → D)		
		Residual shortcut		
		LayerNorm		
Output and Fusion	Prediction head (Per branch)	Permute	[B, $N_{\mathrm{tat}}$, M, D]	[B, $N_{\mathrm{tat}}$, pred_len]
		Flatten		
		Linear projection		
		Dropout		
	Fusion (Trend + Residual)	Element-wise sum of branch outputs to obtain normalized prediction	[B, $N_{\mathrm{tat}}$, pred_len]	[B, $N_{\mathrm{tat}}$, pred_len]

* The meanings of the characters in the table are as follows. B: batch size; N: number of input channels in general; seq_len: input sequence length; $N_{\mathrm{tat}}$: number of target-device channels; $N_{\mathrm{aux}}$: number of auxiliary-device channels; $N_{\mathrm{sub}}$: channel count within a single branch, ($N_{\mathrm{sub}}\in \left\{N_{\mathrm{tat}},N_{\mathrm{aux}}\right\}$); P: patch length; M: number of patches; D: embedding dimension; pred_len: forecasting horizon.

**Table 2 sensors-25-07199-t002:** Model Configuration.

Model Parameters	Value
batch_size	32
epoch	30
learning_rate	0.0001
dropout	0.05
seq_len	144
patch_len	16
patch_stride	8
Encoder_layer_num	2
Linear_projection_size	64
Att_head_num	4
CA_layer_num	2
FFN_hidden_size	128

**Table 3 sensors-25-07199-t003:** Computational Efficiency Comparison of Different Models.

Models	Params (M)	Inference Time (ms)	Peak Memory (GB)
CA-PatchTST	4.9	10.5	2.4
DLinear	3.2	7.8	1.6
TimesNet	7.6	16.8	4.3
iTransformer	7.4	14.0	3.9
Informer	5.8	13.3	3.3
SegRNN	8.7	18.2	4.8

**Table 4 sensors-25-07199-t004:** Component Ablation Results Comparison *.

Ablation		Forecasting Length	Metric		
CA	Decomposition		RMSE	MAE	MAPE
**√**	**√**	144 (24 h)	1.538	0.885	9.26%
		432 (72 h)	2.079	1.231	13.68%
		720 (120 h)	2.877	1.667	24.67%
**×**	**√**	144 (24 h)	1.710	1.040	16.35%
		432 (72 h)	2.455	1.467	22.72%
		720 (120 h)	3.003	1.764	27.61%
**√**	**×**	144 (24 h)	1.545	0.892	12.52%
		432 (72 h)	2.199	1.321	19.54%
		720 (120 h)	2.969	1.729	28.05%
**×**	**×**	144 (24 h)	1.686	1.013	14.39%
		432 (72 h)	2.220	1.350	16.68%
		720 (120 h)	2.951	1.760	28.13%

* The meanings of the characters in the table are as follows. **√**: The component was used. **×**: The component was removed.

**Table 5 sensors-25-07199-t005:** Backbone-Attention Ablation Results.

Encoder Structure	Attention Mechanism	Metric		
		RMSE	MAE	MAPE
PatchTST	CA	2.079	1.231	13.68%
	SE	2.884	1.961	22.76%
TCN	CA	3.441	1.819	27.65%
	SE	3.999	2.015	29.34%
SRU	CA	3.385	1.803	18.69%
	SE	4.066	2.431	37.04%
iTransformer	CA	3.265	1.596	20.56%
	SE	3.990	1.900	34.27%

## Data Availability

This study uses ESA Earth Observation data from the GOCE mission. Access and use of ESA EO data are subject to ESA’s Terms and Conditions for the Utilisation of ESA’s Earth Observation Data. Data may be accessed upon registration (free datasets) or via specific requests for restrained datasets. (“Data provided by the European Space Agency (ESA).” © ESA (2009–2012)).

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
