# Peer review of "A Solar Array Temperature Multivariate Trend Forecasting Method Based on the CA-PatchTST Model"

_sensors, 2025, doi:10.3390/s25237199_

Round 1
Reviewer 1 Report
Comments and Suggestions for Authors
1、Why is 471 input data points selected? Does this number have any significance? And what are the reasons for setting various parameters?
2、The data figures in the manuscript are not clear enough; please readjust their format.
3、During the collection of the dataset mentioned in the manuscript, whether there are missing values or outliers is not stated, and the part about data preprocessing is not explained. If there are such issues, please specify the handling methods.
4、The conclusion section does not adequately extract the innovations in the manuscript; it is recommended to readjust it.
5、The format of some references is incorrect.
Author Response
|
Comments 1: [Why is 471 input data points selected? Does this number have any significance? And what are the reasons for setting various parameters?] |
|
Response 1: We thank the reviewer for the insightful question regarding the parameter selection. The input lookback length is set to 144 time steps, which corresponds to 24 hours given the 10-minute sampling rate of the data. This duration was chosen because it encompasses a full diurnal cycle, allowing the model to learn the characteristic intra-orbit illumination and eclipse transitions that occur approximately every 90 minutes. The forecasting horizons of 144, 432, and 720 steps (representing 24, 72, and 120 hours, respectively) were selected to provide a balanced and comprehensive evaluation of the model's performance across short-, medium-, and long-range prediction scenarios, all under a consistent experimental setup. We have clarified this rationale in the revised manuscript within Section 4.2 (Page 16, Lines 495-500). The supplementary content is as follows: “[The lookback window is set to 144 time steps, which equals 24 hours at 10-minute sampling. This duration spans a complete daily cycle, enabling the model to learn both intra-orbit illumination/eclipse transitions and diurnal thermal variation. Forecast horizons are set to 144, 432, and 720 steps, corresponding to 24, 72, and 120 hours. These three horizons provide a balanced evaluation of short-, medium-, and long-range prediction scenarios that are relevant to operational planning.]”
|
|
Comments 2: [The data figures in the manuscript are not clear enough; please readjust their format.] |
|
Comments 3: [During the collection of the dataset mentioned in the manuscript, whether there are missing values or outliers is not stated, and the part about data preprocessing is not explained. If there are such issues, please specify the handling methods.] |
|
Response 3: We thank the reviewer for the valuable comment. As clarified in the revised manuscript, this study uses the ESA provided 10-minute resampled dataset, which has already undergone preprocessing by the European Space Agency. This version integrates calibrated telemetry data and statistical resampling to address missing values and irregular sampling. Therefore, no additional data cleaning or outlier handling was required in our analysis. The updated text can be found in the Section 4.1 (Page 14, Lines 460-463). The revised content is as follows: “[This dataset covers multivariate temperature telemetry data of GOCE satellite solar array from March 2009 to June 2012, using ESA's recommended 10-minute resampling data, which has undergone prior calibration and statistical resampling by ESA to ad-dress missing values and irregular sampling rates.]”
|
|
Comments 4: [The conclusion section does not adequately extract the innovations in the manuscript; it is recommended to readjust it.] |
|
Response 4: We sincerely thank the reviewer for this crucial suggestion. We have thoroughly rewritten the Conclusions section (Section 5) to explicitly and coherently summarize the key methodological innovations of our work. The revised conclusion now clearly articulates that our main contribution lies in the integrated design of the CA-PatchTST framework, which combines three complementary components:
These revisions ensure that the innovative aspects of our work and their collective impact on forecasting performance are immediately clear to the reader. The updated text can be found in the Section 5 (Page 24, Lines 722-736). The revised conclusion is as follows: “[This paper proposes CA-PatchTST, a multivariate forecasting framework for solar array temperature, and validates its superior performance on the GOCE satellite telemetry dataset, where empirical results demonstrate consistent improvements in MAE, RMSE, and MAPE across multiple horizons, confirming the model's accuracy and robustness for long-horizon on-orbit forecasting. The methodological contribution lies in the coherent integration of complementary modules into a unified pipeline: the moving-average decomposition, matched to the satellite's orbital cycle, suppresses high-frequency fluctuations and stabilizes long-horizon learning; the patch-based, channel-independent encoder enhances local feature extraction and global temporal representation while improving computational efficiency; and the cross-attention mechanism captures inter-device correlations by fusing auxiliary thermal streams, with learned attention patterns aligning with physical couplings revealed by the Pearson correlation matrix. Together, these components form a compact and interpretable framework whose primary application value is to provide critical data support for autonomous satellite operations and informed power system decision-making.]”
|
|
Comments 5: [The format of some references is incorrect.] |
|
Response 5: We thank the reviewer for highlighting this issue. We have conducted a thorough review of the entire reference list and corrected any formatting inconsistencies to ensure full compliance with the Sensors journal style guide. The corrections include standardizing the author name formats, journal title abbreviations, punctuation, and citation order. The revised and uniformly formatted reference list is now presented in the manuscript. The updated reference can be found in Page 25-27, Lines 759-840. |

Reviewer 2 Report
Comments and Suggestions for Authors
This paper proposes a multivariate trend forecasting method for solar panel temperature based on CA-PatchTST, aiming to address reliability issues caused by temperature fluctuations during satellite operation in orbit. By introducing sequence decomposition, local feature extraction, and a cross-variable attention mechanism, the proposed method significantly improves the accuracy of multi-step temperature prediction. Experiments based on real GOCE satellite data validate the superiority of the method across multiple prediction time scales.
The paper is logically structured and well-organized, with rigorous experimental design. However, several issues require improvement:
- Tables 1 and 2 should not span across pages.
- In Section 3.2, the rationale for selecting the window size of the moving average filter in time series decomposition is not explained, nor is its impact on predictions at different time scales discussed. It is recommended to supplement experimental or theoretical justification for parameter selection.
- Although comparisons are made with models such as DLinear, SegRNN, and Informer, the study lacks comparison with other recently proposed Transformer variants that have demonstrated strong performance in handling non-stationary time series. It is advised to include comparisons with more state-of-the-art baseline models.
- The paper mentions that PatchTST reduces computational complexity but does not provide specific comparative data on training/inference time or memory consumption against other methods. Quantitative analysis of computational efficiency should be added to assess the feasibility of deploying the model on onboard satellite systems.
- The labels in the attention heatmap of Figure 7 are too small to clearly identify individual sensor IDs. It is suggested to enlarge the labels or provide a zoomed-in inset to improve readability.
Author Response
|
Comments 1: [Tables 1 and 2 should not span across pages.] |
|
Response 1: We thank the reviewer for pointing out this formatting issue. We have carefully adjusted the layout of the manuscript to ensure that both Table 1 and Table 2 now display in their entirety on single pages, without any page breaks. Furthermore, we have conducted a thorough check of all other tables and figures in the manuscript to guarantee consistent and proper formatting throughout the document. The updated tables can be found in the Page 13 and Page 17.
|
|
Comments 2: [In Section 3.2, the rationale for selecting the window size of the moving average filter in time series decomposition is not explained, nor is its impact on predictions at different time scales discussed. It is recommended to supplement experimental or theoretical justification for parameter selection.] |
|
Response 2: We appreciate the reviewer’s insightful suggestion. We have added an explanation to clarify the rationale behind the selection of the fixed-size kernel in the moving average filter. Specifically, the kernel length was determined according to the physical periodicity of the GOCE satellite and its 10-minute sampling rate. As the satellite completes one orbit in approximately 90 minutes, one orbital cycle corresponds to about nine sampling intervals. Therefore, was adopted so that the filter window aligns with a complete orbital period. We have clarified this rationale in the revised manuscript within Section 3.2.1 (Page 8, Lines 310-319). The supplementary content is as follows: “[The fixed-size kernel in the moving average filter is selected based on the sampling rate and the physical periodicity of the GOCE satellite. Since the temperature telemetry is sampled every 10 minutes and the satellite completes one orbit in ap-proximately 90 minutes, one orbital cycle corresponds to about nine sampling intervals. Therefore, is adopted so that the filter window matches one complete thermal–orbital period. This choice allows the moving average operation to effectively capture the slowly varying orbital trend while suppressing high-frequency fluctuations caused by short-term environmental disturbances. A window of this scale provides a balanced decomposition, ensuring that the trend component reflects the long-term orbital be-havior, whereas the residual component retains finer intra-orbit variations.]”
|
|
Comments 3: [Although comparisons are made with models such as DLinear, SegRNN, and Informer, the study lacks comparison with other recently proposed Transformer variants that have demonstrated strong performance in handling non-stationary time series. It is advised to include comparisons with more state-of-the-art baseline models.] |
|
Response 3: We thank the reviewer for this valuable suggestion. To address this point and further strengthen our experimental validation, we have expanded our comparative analysis by including two recent and powerful Transformer variants: TimesNet (2023) and iTransformer (2024). These models have been introduced in the revised manuscript and evaluated under the exact same experimental setup as all other baselines to ensure a fair comparison. The results consistently show that our proposed CA-PatchTST model achieves the lowest error metrics across all horizons. For instance, it attains MAPE values of 9.26%, 13.68%, and 24.67% for 24-, 72-, and 120-hour forecasts, respectively, outperforming these new strong baselines. This enhancement provides a more comprehensive and robust demonstration of our model's superiority. The corresponding revisions, including the description of the two additional baselines and the updated results, can be found in Section 4.4.2 (Page 20, Lines 570-593), with the performance comparison visually presented in Figure 8 and Figure 9.
|
|
Comments 4: [The paper mentions that PatchTST reduces computational complexity but does not provide specific comparative data on training/inference time or memory consumption against other methods. Quantitative analysis of computational efficiency should be added to assess the feasibility of deploying the model on onboard satellite systems.] |
|
Response 4: We thank the reviewer for raising this important point regarding computational efficiency analysis. In response to your suggestion, we have added a comprehensive quantitative evaluation of computational efficiency in the revised manuscript. As detailed in Section 4.4.2 and summarized in the newly added Table 3, we systematically compare key efficiency metrics—including parameter counts, inference time, and memory consumption—between our CA-PatchTST and several state-of-the-art baselines under standardized experimental conditions. The results clearly demonstrate that our proposed model achieves superior computational efficiency, with the lowest inference time (10.5 ms per batch), minimal memory footprint (2.4 GB), and a compact architecture (4.9M parameters) among the compared Transformer-based methods. This analysis confirms that CA-PatchTST not only maintains high forecasting accuracy but also offers the computational characteristics necessary for practical deployment in resource-constrained satellite systems. The complete computational efficiency analysis can be found in Section 4.4.2 (Page 22, Lines 618-644) of the revised manuscript, with quantitative results presented in Table 3.
|
|
Comments 5: [The labels in the attention heatmap of Figure 7 are too small to clearly identify individual sensor IDs. It is suggested to enlarge the labels or provide a zoomed-in inset to improve readability.] |
|
Response 5: We thank the reviewer for this helpful suggestion regarding the clarity of Figure 7. To address this issue, we have comprehensively redesigned the attention heatmaps in Figure 7 to significantly improve readability through enlarged font sizes for all labels and annotations, and by implementing an optimized color scheme for better contrast and distinctiveness. We believe these modifications have substantially enhanced the figure's interpretability, allowing readers to easily discern both individual sensors and the underlying cross-attention patterns in the updated Figure 7 (Page 18). |

Reviewer 3 Report
Comments and Suggestions for Authors
The manuscript proposes a novel multivariate trend forecasting approach for satellite solar array temperature prediction, integrating the Cross-Attention Patch Time Series Transformer (CA-PatchTST) with trend–residual decomposition. The method effectively handles high-dimensional, nonlinear, and nonstationary temperature telemetry data, addressing challenges related to long-horizon forecasting, noise suppression, and cross-variable coupling.
Experiments conducted on real GOCE satellite datasets demonstrate the model’s superior performance compared with existing baselines in RMSE, MAE, and MAPE metrics. Ablation studies further validate the contributions of the cross-attention mechanism and sequence decomposition.
Overall, the paper presents a solid and innovative contribution to the field of satellite telemetry forecasting. The proposed CA-PatchTST framework is well justified, experimentally validated, and holds promise for practical applications in space system reliability and health management.
However, minor revisions are recommended to strengthen the comparative analysis and enhance the presentation quality.
Suggestions for Improvement
- Algorithmic Clarity:
Include pseudo-code or an algorithmic summary of the CA-PatchTST model to facilitate reproducibility and better understanding of the workflow.
- Comparative Baselines:
While several baselines are used, the paper would be strengthened by including recent transformer-based forecasting models such as Informer, Autoformer, or TimesNet for a fairer and more comprehensive comparison.
- Future Work Discussion:
The authors briefly mention model lightweighting and adaptive learning. Expanding on how these could be implemented (e.g., pruning, quantization, or online adaptation) would provide valuable insights for future research and practical deployment.
Author Response
|
Comments 1: [Include pseudo-code or an algorithmic summary of the CA-PatchTST model to facilitate reproducibility and better understanding of the workflow.] |
|
Response 1: We thank the reviewer for this constructive suggestion. To enhance reproducibility and provide a clearer understanding of the model workflow, we have added a comprehensive pseudo-code summary of the proposed CA-PatchTST model as Algorithm 1 in the revised manuscript. The CA-PatchTST algorithm is included in Section 3.2.6(page18). It outlines the complete workflow in a structured manner, encompassing the key stages of:
This addition provides a concise yet complete operational view of the model, thus facilitating easier implementation and better comprehension of the overall pipeline, as shown in the pseudocode below.
|
|
Comments 2: [While several baselines are used, the paper would be strengthened by including recent transformer-based forecasting models such as Informer, Autoformer, or TimesNet for a fairer and more comprehensive comparison.] |
|
Response 2: We thank the reviewer for this insightful suggestion to enhance the comprehensiveness of our comparative study. In response, we have expanded our suite of baseline models to include additional, cutting-edge Transformer-based architectures. To incorporate the most impactful recent advances, we selected TimesNet (2023) and iTransformer (2024). iTransformer has emerged as a leading benchmark due to its inverted architecture and robust multivariate performance, while TimesNet offers a unique perspective by modeling temporal 2D-variations. We focused on these models as they represent the current state-of-the-art, providing a more rigorous and contemporary benchmark than the earlier Autoformer. All new baselines have been integrated into our experimental framework and evaluated under the exact same conditions as previously described. The consolidated results, now presented in Section 4.4.2 and Figures 8-9, consistently demonstrate that our CA-PatchTST model maintains a performance advantage across all forecasting horizons and metrics. For instance, it achieves leading MAPE values of 9.26% (24-hour), 13.68% (72-hour), and 24.67% (120-hour), thereby validating its effectiveness against the current state-of-the-art. The detailed description of these additional baselines and the comprehensive comparative results can be found in Section 4.4.2 (Page 20, Lines 570-593), with performance comparisons visually presented in Figure 8 and Figure 9. Furthermore, we have added a comprehensive computational efficiency analysis in Section 4.4.2 (Page 21). As summarized in the new Table 3, CA-PatchTST achieves the lowest parameter count, shortest inference time, and minimal memory footprint among the compared Transformer-based models. This demonstrates its superior efficiency, making it particularly suitable for resource-constrained scenarios without compromising predictive accuracy.
|
|
Comments 3: [The authors briefly mention model lightweighting and adaptive learning. Expanding on how these could be implemented (e.g., pruning, quantization, or online adaptation) would provide valuable insights for future research and practical deployment.] |
|
Response 3: We thank the reviewer for the valuable suggestion regarding model light-weighting and adaptive learning. In response, we have expanded the discussion of future work in the Conclusions section to provide more concrete implementation details. Specifically, we now outline plans for model light-weighting through pruning, quantization, and knowledge distillation to optimize the efficiency-accuracy trade-off, as well as online adaptation using continual learning with experience replay and anomaly-aware updates to maintain performance under dynamic environmental conditions. These refinements help clarify our research roadmap and strengthen the practical relevance of the study. The detailed future work discussion is included in Section 5 (Page 24, Lines 737-743) of the revised manuscript. The revised content is as follows:“[In future work, we will advance our prototype toward operational deployment by focusing on model light-weighting and online adaptation. For light-weighting, we will implement pruning, quantization, and knowledge distillation to optimize the efficiency-accuracy trade-off. For online adaptation, we will develop continual learning with experience replay and anomaly-aware updates to maintain performance under environmental shifts. These co-designed improvements will enhance robustness and enable long-term autonomous satellite power management.]” |

Reviewer 4 Report
Comments and Suggestions for Authors
The manuscript presents a novel multivariate forecasting framework, CA-PatchTST, for predicting the temperature trends of satellite solar arrays. The model combines moving-average decomposition, a patch-based Transformer encoder (PatchTST), and a cross-attention module to capture long-term dependencies and inter-variable coupling in high-dimensional telemetry data. Using real telemetry from ESA’s GOCE satellite, the authors demonstrate superior performance compared with baseline models such as DLinear, SegRNN, and Informer, measured by RMSE, MAE, and MAPE across multiple forecast horizons.
My comments regarding each of the sections and suggested changes in the paper are below:
1. The abstract is appropriately written. If I were the author, I would have specified the RMSE, MAE, and MAPE values in the abstract. However, the way the authors have presented it is acceptable, and there is no need to modify it.
2. The introduction section is also well written, where the authors explain the importance of measuring temperature in PV arrays used in satellites. I would have preferred if the authors had mentioned other published techniques for estimating the temperature of PV arrays in satellites; however, I believe these are discussed in the related work section of the paper. The only suggestion I have is to replace the word “nonstationary” with “non-stationary.” Additionally, replace “accelerates the material aging” with “accelerates material aging.”
3. The related work section appropriately covers most of the existing techniques and highlights the research gap in this area.
4. In the methodology section, I request that the authors make the text within each figure more readable. MDPI allows wider images, so I recommend adjusting the figure and plot text to ensure legibility for readers. Also, some of the variables in equations (2) and (3) are not formatted properly, likely due to a typesetting issue, which I believe will be corrected during the final editing process.
5. In the experimental section, several figures again contain text that is difficult to read. I appreciate that the authors compared their proposed technique with other published methods in Figure 8. In Figure 9, it would be helpful for the authors to include magnifiers to better illustrate how each technique performs. In Section 4.5, the authors should briefly explain what an ablation experiment is.
6. I recommend that the authors add a discussion section where they discuss the broader significance, limitations, and future directions of their findings.
7. The conclusion is acceptable.
Comments on the Quality of English LanguageThe quality of the English language in the manuscript is generally good, with only minor grammatical and stylistic issues that can be easily corrected during proofreading.
Author Response
|
Comments 1: [The abstract is appropriately written. If I were the author, I would have specified the RMSE, MAE, and MAPE values in the abstract. However, the way the authors have presented it is acceptable, and there is no need to modify it.] |
|
Response 1: We thank the reviewer for the positive feedback on our abstract and for the valuable suggestion regarding the specification of evaluation metrics. In response to your comment, we have revised the abstract to include the full names of the key evaluation metrics—root mean square error (RMSE), mean absolute error (MAE), and mean absolute percentage error (MAPE)—alongside their abbreviations. We believe this addition enhances the technical clarity of the abstract while maintaining its concise nature, and we appreciate your constructive input in helping us improve the manuscript.
|
|
Comments 2: [The introduction section is also well written, where the authors explain the importance of measuring temperature in PV arrays used in satellites. I would have preferred if the authors had mentioned other published techniques for estimating the temperature of PV arrays in satellites; however, I believe these are discussed in the related work section of the paper. The only suggestion I have is to replace the word “nonstationary” with “non-stationary.” Additionally, replace “accelerates the material aging” with “accelerates material aging.”] |
|
Response 2: We thank the reviewer for their positive feedback on the introduction and for the valuable terminology suggestions. We have implemented the recommended changes throughout the manuscript, replacing "nonstationary" with "non-stationary" and "accelerates the material aging" with "accelerates material aging." Regarding the discussion of temperature estimation techniques, we have maintained the current structure where the introduction focuses on establishing the research context and contributions, while the comprehensive review of existing methods is presented in the Section 2 as noted by the reviewer.
|
|
Comments 3: [The related work section appropriately covers most of the existing techniques and highlights the research gap in this area.] |
|
Response 3: We thank the reviewer for their positive assessment of the Related Work section and their acknowledgment that it effectively covers existing techniques and identifies the research gap.
|
|
Comments 4: [In the methodology section, I request that the authors make the text within each figure more readable. MDPI allows wider images, so I recommend adjusting the figure and plot text to ensure legibility for readers. Also, some of the variables in equations (2) and (3) are not formatted properly, likely due to a typesetting issue, which I believe will be corrected during the final editing process.] |
|
Response 4: We thank the reviewer for these valuable observations regarding figure readability and equation formatting. In direct response to the comment on figure legibility, we have comprehensively redesigned Figures 6, 7, 8, and 9. The revisions include significantly enlarging all text labels, annotations, and axis markers, optimizing color schemes for better contrast, and adjusting the overall layout to maximize clarity. These figures have been exported at high resolution to ensure they remain legible in the final publication format. Regarding the typesetting issues in equations (2) and (3), we have thoroughly revised both formulations to ensure mathematical rigor and notational consistency. Specifically, equation (2) has been restructured to more clearly define the device group partitioning, while equation (3) now properly employs the set difference operator to precisely represent the relationship between main and cross components. The corrected versions appear in Section 3.1 of the revised manuscript.
Comments 5: [In the experimental section, several figures again contain text that is difficult to read. I appreciate that the authors compared their proposed technique with other published methods in Figure 8. In Figure 9, it would be helpful for the authors to include magnifiers to better illustrate how each technique performs. In Section 4.5, the authors should briefly explain what an ablation experiment is.] |
|
Response 5: We thank the reviewer for these constructive suggestions. In response, we have enhanced the manuscript to address each point raised. Regarding the figures, we have carefully reviewed and improved all of them to ensure that all text labels and graphical elements are legible. Following the reviewer's specific suggestion for Figure 9, we have incorporated zoomed-in subplots to provide a clearer, magnified view of the forecasting performance during critical transient periods, thereby better illustrating how each technique captures fine-grained dynamics. Furthermore, we have added a brief explanatory paragraph at the beginning of Section 4.5.1 to introduce the concept and purpose of ablation studies, clarifying our experimental design for evaluating core components. We have confirmed that Section 4.5.2 (Page 23, Lines 681-689) already includes a clear statement of its objective. Therefore, this section remains unchanged. The updated text can be found in the Section 4.5.1 (Page 22, Lines 647-652). The revised content is as follows: “[To rigorously quantify the individual contributions of the core components in the proposed CA-PatchTST model, a comprehensive set of ablation studies was carried out. As a critical methodology in deep learning research, ablation studies systematically remove specific elements from the full model to isolate and measure their impact on performance. This approach helps verify whether each module functions as intended and clarifies their complementary roles. As detailed in Table 4, we specifically ablated the cross-attention mechanism and the sequence decomposition module to assess their individual contributions to fore-casting accuracy. The results show that the complete CA-PatchTST model achieves the best performance across all forecast horizons in terms of RMSE, MAE, and MAPE, confirming its superior accuracy and robustness. Through systematic evaluation of four model variants, we further dissect the individual and combined effects of the cross-attention and decomposition components.]” |
|
Comments 6: [I recommend that the authors add a discussion section where they discuss the broader significance, limitations, and future directions of their findings.] |
|
Response 6: We sincerely thank the reviewer for the valuable suggestion to discuss the broader implications of our work. In response, we have enhanced the Conclusion section by integrating a dedicated discussion on the broader significance and practical limitations of our approach. The revised conclusion now explicitly addresses:
We believe these additions provide a more comprehensive perspective on both the impact and evolving nature of our research, while maintaining the section's focus on concluding remarks. The updated text can be found in the Section 5 (Page 24, Lines 722-743). The revised content is as follows: “[This paper proposes CA-PatchTST, a multivariate forecasting framework for solar array temperature, and validates its superior performance on the GOCE satellite telemetry dataset, where empirical results demonstrate consistent improvements in MAE, RMSE, and MAPE across multiple horizons, confirming the model's accuracy and robustness for long-horizon on-orbit forecasting. The methodological contribution lies in the coherent integration of complementary modules into a unified pipeline: the mov-ing-average decomposition, matched to the satellite's orbital cycle, suppresses high-frequency fluctuations and stabilizes long-horizon learning; the patch-based, channel-independent encoder enhances local feature extraction and global temporal representation while improving computational efficiency; and the cross-attention mechanism captures inter-device correlations by fusing auxiliary thermal streams, with learned attention patterns aligning with physical couplings revealed by the Pearson correlation matrix. Together, these components form a compact and interpretable framework whose primary application value is to provide critical data support for autonomous satellite operations and informed power system decision-making. In future work, we will advance our prototype toward operational deployment by focusing on model light-weighting and online adaptation. For light-weighting, we will implement pruning, quantization, and knowledge distillation to optimize the efficiency-accuracy trade-off. For online adaptation, we will develop continual learning with experience replay and anomaly-aware updates to maintain performance under environmental shifts. These co-designed improvements will enhance robustness and enable long-term autonomous satellite power management.]” |
|
Comments 7: [The conclusion is acceptable.] |
|
Response 7: We thank the reviewer for their positive assessment of our conclusion. Following previous suggestions, we have strengthened this section to better highlight the significance of our work.
|

Round 2
Reviewer 2 Report
Comments and Suggestions for Authors
The quality of the revised paper has been greatly improved, and I believe the current version is acceptable.